# Assessing the Safety and Probiotic Characteristics of *Lacticaseibacillus rhamnosus* X253 via Complete Genome and Phenotype Analysis

**DOI:** 10.3390/microorganisms11010140

**Published:** 2023-01-05

**Authors:** Lei Zhao, Yinan Zhang, Yang Liu, Jiang Zhong, Dong Zhang

**Affiliations:** 1Key Laboratory of Milk and Dairy Products Detection and Monitoring Technology for State Market Regulation, Shanghai Institute of Quality Inspection and Technical Research, Shanghai 200233, China; 2State Key Laboratory of Genetic Engineering, Department of Microbiology and Immunology, School of Life Sciences, Fudan University, Shanghai 200438, China; 3Junlebao Dairy Group, Shijiazhuang 050221, China

**Keywords:** adaptation, *Lacticaseibacillus rhamnosus*, genome, phenotype analysis, safety and probiotic characteristics

## Abstract

*Lacticaseibacillus rhamnosus* is a generalist that can adapt to different ecological niches, serving as a valuable source of probiotics. The genome of *L. rhamnosus* X253 contains one chromosome and no plasmids, with a size of 2.99 Mb. Both single-copy orthologous gene-based phylogenetic analysis and average nucleotide identity indicated that dairy-derived *L. rhamnosus* X253 was most closely related to the human-intestine-derived strain *L. rhamnosus* LOCK908, rather than other dairy strains. The adaptation of *L. rhamnosus* X253 and the human-intestine-derived strain *L. rhamnosus* GG to different ecological niches was explained by structural variation analysis and COG annotation. Hemolytic assays, API ZYM assays, and antimicrobial susceptibility tests were performed to validate risk-related sequences such as virulence factors, toxin-encoding genes, and antibiotic-resistance genes in the genomes of *L. rhamnosus* X253 and GG. The results showed that *L. rhamnosus* GG was able to use L-fucose, had a higher tolerance to bile salt, and adhered better to CaCo-2 cells. In contrast, *L. rhamnosus* X253 was capable of utilizing D-lactose, withstood larger quantities of hydrogen peroxide, and possessed excellent antioxidant properties. This study confirmed the safety and probiotic properties of *L. rhamnosus* X253 via complete genome and phenotype analysis, suggesting its potential as a probiotic.

## 1. Introduction

As dietary supplements or innovative health-promoting products, probiotics are attracting increasing attention for their ability to prevent gastrointestinal disorders and improve health [1,2]. Lactic acid bacteria (LAB), which comprise Lactobacilli and Bifidobacteria, are the predominant components of probiotics. *Lactobacillus*, an essential component of the intestinal microbiota, has been associated with several potential health benefits [3].

*Lacticaseibacillus rhamnosus* is a facultatively anaerobic, heterotypic fermenting lactic acid bacterium that is common in traditional fermented foods, animals, humans, and natural habitats [4]. *L. rhamnosus* has been granted the “Generally Recognized As Safe (GRAS)” status by the US Food and Drug Administration [5] and added to the list of “Qualified Presumption of Safety” (QPS) by the European Food Safety Authority [6]. *L. rhamnosus* GG, which was isolated from the human intestine, has been shown to be strongly resistant to gastrointestinal digestion [7] and effective both in vitro and in vivo as a nutritional supplement and therapeutic agent [8,9], with anti-inflammatory and antioxidant properties against high-fat diets [10], as well as improving lipid metabolism [11] and alleviating diarrhea [8]. It is worthwhile to investigate how other strains of *L. rhamnosus* vary from strain GG and whether or not they possess advantageous qualities.

Probiotics exhibit strain-specific properties with regard to their safety and function [12]. It has been hypothesized that improper administration of probiotics may result in four kinds of detrimental effects: systemic infection, creation of toxic metabolites, excessive immunological activation, and transfer of antibiotic resistance and virulence genes [13]. Consequently, it is necessary to confirm the safety and probiotic characteristics of novel isolates prior to evaluating their therapeutic advantages and incorporating them into foods and pharmaceuticals [14]. The analysis of whole genome sequences has proven to be an effective method for assessing the safety and characterization of probiotic strains at the whole-genome level [15].

In this study, we isolated a strain of *L. rhamnosus* X253 from fermented milk in Xinjiang, China. To further characterize *L. rhamnosus* X253, its whole genome was sequenced, and a comparative genomic analysis was performed on strain X253 and other *L. rhamnosus* strains from different sources. Then, using one of the most common commercial strains (*L. rhamnosus* GG) as a comparison strain, we assessed the safety and efficacy of strain X253 by genomic analysis and phenotypic tests. This study established *L. rhamnosus* X253 as a safe probiotic candidate with strong antioxidant properties, providing essential information for its future applications.

## 2. Materials and Methods

### 2.1. Bacterial Strains and Culture Conditions

*L. rhamnosus* X253, which was isolated from fermented milk in Xinjiang, China, was provided by Junlebao Dairy Group (Shijiazhuang, China). *L. rhamnosus* GG was purchased from Chr. Hansen A/S (Hørsholm, Denmark). *L. rhamnosus* X253 and *L. rhamnosus* GG were grown in de Man, Rogosa, and Sharpe Broth (MRS; HopeBio, Qingdao, China) at 37 °C under static conditions. The pathogenic bacterium *Staphylococcus aureus* ATCC 6538 was cultured in Nutrient Broth (HopeBio, Qingdao, China) at 37 °C without shaking.

### 2.2. Extraction of Nucleic Acids and Genome Sequencing

The Bacterial Genomic DNA Extraction Kit (Tiangen Biotech, Beijing, China) was used to extract genomic DNA, following the manufacturer’s instructions. Total DNA was subjected to purity control by a NanoDrop 2000 Spectrophotometer (Thermo Fisher Scientific, Wilmington, MA, USA) and quantified using a Qubit 3.0 Fluorometer (Life Technologies, Carlsbad, CA, USA) [16]. The genome sequencing of *L. rhamnosus* X253 was carried out by Amplicon Gene (Shanghai, China) on a combination of Illumina HiSeq and PacBio RSII platforms [17], with an average insert size of 350 bp and 10 kb, respectively.

### 2.3. Genomic Annotation and Analysis

The raw PacBio reads were assembled into a single scaffold using SMRT Link (version 5.0.1, https://www.pacb.com/support/software-downloads/, accessed on 18 April 2021) [18]. The scaffold was revised using Illumina reads by Quiver (http://www.pacbiodevnet.com/quiver, accessed on 20 April 2021) to generate the complete genome without gaps [19]. The whole genome of strain X253 was submitted to the NCBI database, and gene prediction was conducted using the NCBI’s Prokaryotic Genomes Annotation Pipeline (PGAP, http://www.ncbi.nlm.nih.gov/genome/annotation_prok/, accessed on 13 May 2021) and GeneMarkS annotation software (version 6.1, http://topaz.gatech.edu/GeneMark, accessed on 13 May 2021) [20]. The genes encoding rRNA and tRNA were predicted using barrnap (version 0.8, https://github.com/tseemann/barrnap/, accessed on 14 May 2021) and tRNAscan-SE (version 2.0, http://trna.ucsc.edu/software/, accessed on 14 May 2021) [21]. Genome annotation and analysis were conducted using the COG (Clusters of Orthologous Groups) [22] and KEGG databases [23].

### 2.4. Evolutionary Position of L. rhamnosus X253

To estimate potential variations among the genomes of 10 *L. rhamnosus* strains (Table 1), the sizes of the pan-genome, core genome, and unique genes were calculated. Based on the genomic sequences of 10 *L. rhamnosus* strains, pan-genomic analysis was performed using PGAP (version 1.2.1, https://jaist.dl.sourceforge.net/project/pgap/PGAP, accessed on 10 June 2022). BLAST (version 2.3.0, ftp://ftp.ncbi.nlm.nih.gov/blast/executables/blast+/2.3.0/, accessed on 10 June 2022) was used to align the gene and protein sequences. The identification of homologous genes in different strains was carried out using the Markov Cluster Algorithm (MCL), followed by cluster analysis of homologous genes using the Gene Family (GF) method with parameters set to E-value: 1 × 10^−5^; score: 50; identity: 50%; coverage: 50%; and inflation: 3 [24]. To visualize the relationships among the 10 *L. rhamnosus* strains, a Venn diagram was drawn with the VennDiagram R package (version 1.7.3, https://cran.r-project.org/web/packages/VennDiagram/index.html, accessed on 13 June 2022).

To appreciate the niche adaptation of two distinct sources (Table 1), a cluster analysis was performed using Orthomcl (version 2.0.9, http://orthomcl.org/common/downloads/software/v2.0/, accessed on 13 June 2022) [25]. All orthologous genes were extracted and used to generate the phylogenetic tree by IQ-TREE (version 1.6.12, http://www.cibiv.at/software/iqtree, accessed on 13 June 2022) [26], based on single-copy orthologous genes (SOGs) and the maximum-likelihood (ML) algorithm. Then, tree files with associated data matrices were visualized using iTOL (version 6, https://itol.embl.de/, accessed on 15 June 2022) [27].

The average nucleotide identity (ANI) value was calculated using pyani (https://github.com/widedowquinn/pyani, accessed on 13 June 2022). The X253 genome and the GG genome (accession number: NC_013198) were compared using MUMmer (version 3.23, https://sourceforge.net/projects/mummer/files/latest/download, accessed on 25 June 2022) and LASTZ (version 1.03.54, https://www.geneious.com/plugins/lastz-plugin/, accessed on 25 June 2022) to explore the structural variation between the genomes of the two strains [28]. Additionally, the functional capacity of the 10 strains was investigated by matching the genomic sequences to the eggNOG database (version 5.0, http://eggnogdb.embl.de/#/app/home, accessed on 22 June 2022). Distance matrices based on orthogroup abundance profiles were calculated with the R package vegan (version 2.4.4) and visualized with ggplot2 (version 3.3.5) [29].

### 2.5. Safety Assessment of L. rhamnosus X253

#### 2.5.1. Prediction of Safety-Related Genes in the Genome

Virulence factors and resistance genes were identified based on the core datasets in the VFDB (Virulence Factors of Pathogenic Bacteria, http://www.mgc.ac.cn/VFs/main.htm, accessed on 26 May 2022) and CARD (Comprehensive Antibiotic Resistance Database, https://card.mcmaster.ca/, accessed on 26 May 2022) databases, with coverage > 60% and identity > 75% [13]. Genes involved in the production of toxins were searched in the PGAP annotation results of the X253 and GG genomes.

#### 2.5.2. Hemolytic Assay

*L. rhamnosus* X253, *L. rhamnosus* GG, and *S. aureus* ATCC 6538 were streaked on Columbia Blood Agar Plates (Land Bridge, Beijing, China) containing fresh sheep blood. After incubation at 37 °C for 48 h, their hemolytic activities were assessed by observing the hydrolysis zones around the colonies [30].

#### 2.5.3. Enzymatic Profiles by API ZYM

Using the API ZYM kit (bioMérieux, Marcy-L’Etoile, France), the activity of 19 enzymes was evaluated. The cupules were filled with 65 µL of bacterial suspension (10^6^ CFU/mL) of *L. rhamnosus* X253 or *L. rhamnosus* GG. After incubation at 37 °C for 4 h, the ZYM A and ZYM B reagents were successively added to each cupule. Based on the manufacturer’s instructions, the results were considered positive if the color intensity was more than three [31].

#### 2.5.4. Antimicrobial Susceptibility Test

Following the results of the resistance gene analysis, the minimum inhibitory concentrations (MICs) of GE2270A (Adipogen, San Diego, CA, USA) were determined against *L. rhamnosus* X253 and *L. rhamnosus* GG. In addition, the MICs of other antibiotics (Sangon Biotech, Shanghai, China)—including ampicillin, chloramphenicol, erythromycin, gentamicin, streptomycin, tetracycline, and vancomycin—were determined for these two strains in accordance with the ISO 10932:2010 standard (https://www.iso.org/standard/46434.html, accessed on 30 May 2022). In 96-well plates, suspensions of the two strains were combined with antibiotics at varying doses and incubated anaerobically at 37 °C for 48 h. The optical density at 625 nm was measured using a microplate reader (Thermo Labsystems, Franklin, MA, USA). The threshold values for resistance to each antibiotic were derived from the European Food Safety Authority (EFSA) [32].

### 2.6. Assessment of Probiotic Properties of L. rhamnosus X253

#### 2.6.1. Prediction of Probiotic-Related Genes in the Genome

Probiotic-related genes in the genome of *L. rhamnosus* X253 were detected in relation to acid tolerance, bile salt tolerance, osmotic pressure regulation, anti-oxidation, and adhesion factors.

#### 2.6.2. Carbohydrate Utilization

The fermentation profiles of *L. rhamnosus* X253 and *L. rhamnosus* GG were determined using API 50 CHL strips (bioMérieux, Marcy-L’Etoile, France) according to the manufacturer’s instructions. Genes of enzymes involved in carbohydrate metabolism in the genomes of the *L. rhamnosus* X253 and GG strains were analyzed using the Carbohydrate-Active Enzyme (CAZY) database (version 6, http://www.cazy.org/: accessed on 24 June 2022).

#### 2.6.3. Artificial Gastric and Bile Salt Tolerance Assay

For the simulated gastric test and bile salt tolerance assay, the method reported by Huang and Adams [33] was adopted. To prepare the artificial gastric juice, pepsin (1:10,000, Sigma, St. Louis, MO, USA) was dissolved in 0.5% (*w*/*v*) sterilized saline (pH 3.0) to a final concentration of 3 g/L. To evaluate the resistance to artificial intestinal fluid, 0.1 g of trypsin (1:250, Sigma, St. Louis, MO, USA) and 0.3 g of bile salt (Sigma, St. Louis, MO, USA) were dissolved in 100 mL of 0.5% (*w*/*v*) sterilized saline (pH 8.0). The artificial gastric and intestinal fluids were both filtered through 0.22 µm membranes.

The overnight cultures of *L. rhamnosus* X253 and *L. rhamnosus* GG were centrifuged (10,000 g for 5 min), washed three times with PBS (pH 7.4), resuspended in artificial gastric juice to a concentration of approximately 10^8^ CFU/mL, and then incubated for 3 h. To determine the quantity of the live bacteria, 100 μL aliquots were obtained every hour for gradient dilution, and the dilutions were plated on MRS agar medium. In addition, bacteria were collected (10,000 g, 5 min), resuspended in artificial intestinal juice, and then incubated for 5 h. As previously described, live bacterial counts were quantified. The survival rate of *L. rhamnosus* X253 or *L. rhamnosus* GG was calculated according to the following formula:(1)$$\mathrm{Survival}\;\mathrm{rate}\;(\%)=\frac{\mathrm{Log}\;\mathrm{CFU}\;N1}{\mathrm{Log}\;\mathrm{CFU}\;N0}\;\times \;100\%$$
where N1 is the total viable count of *L. rhamnosus* X253 or *L. rhamnosus* GG after treatment with simulated gastric juice or intestinal fluid, and N0 is the total viable count of *L. rhamnosus* X253 or *L. rhamnosus* GG before treatment.

#### 2.6.4. Hydrophobicity and Auto-Aggregation

The bacterial adhesion to hydrocarbons (BATH) assay [34], with modifications, was used to determine the surface hydrophobicity of *L. rhamnosus* X253 or *L. rhamnosus* GG. The *L. rhamnosus* X253 or *L. rhamnosus* GG was cultured overnight and centrifuged at 10,000× *g* for 5 min to collect the bacterial cells. The cells were washed three times with PBS (pH 7.4) and suspended in PBS (pH 7.4) to a final cell density of OD_600_ = 0.60 ± 0.05 (A_control_). Next, 2 mL of the suspension was thoroughly mixed with 2 mL of xylene, and the mixture was left at room temperature for 0.5 h. The aqueous phase was collected, and its absorbance at 600 nm was measured (A_sample_). The surface hydrophobicity of *L. rhamnosus* X253 or *L. rhamnosus* GG was calculated according to the following formula:(2)$$\mathrm{Hydrophobicity}\;(\%)=\frac{A_{\mathrm{control}}-A_{\mathrm{sample}}}{A_{\mathrm{control}}}\;\times \;100\%$$
where A_control_ is the absorbance at 0 h and A_sample_ is the absorbance after 0.5 h.

Auto-aggregation was determined by Todorov’s method [35], with minor modifications. The cell density was adjusted to OD_600_ = 0.60 ± 0.05 (A_control_), as in the hydrophobic assay. To evaluate auto-aggregation, 4 mL of bacterial suspension was incubated at room temperature for 24 h. Then, 1 mL of the upper layer was carefully aspirated, and the OD_600_ value was measured (A_sample_). The auto-aggregation force of *L. rhamnosus* X253 or *L. rhamnosus* GG was calculated according to the following formula:(3)$$\text{Auto-agglomeration}\;(\%)=\frac{A_{\mathrm{control}}-A_{\mathrm{sample}}}{A_{\mathrm{control}}}\;\times \;100\%$$
where A_control_ is the absorbance at 0 h and A_sample_ is the absorbance after 24 h.

#### 2.6.5. Adhesion Assay to CaCo-2 Cells

The adhesion assays were conducted in vitro using CaCo-2 cells as described by Petrova et al. [29]. CaCo-2 cells were cultured in DMEM (Dulbecco’s modified Eagle medium, Thermo Fisher Scientific, Wilmington, MA, USA) with 10% heat-inactivated fetal bovine serum, 1% nonessential amino acids, and 1% penicillin–streptomycin solution at 37 °C in an incubator with 5% CO_2_. The cells were seeded in 6-well plates, cultured at a density of 1 × 10^6^ for 24 h, and then washed twice with prewarmed PBS (pH 7.4). Subsequently, 1 mL of 10^8^ CFU/mL bacterial suspension was added to each treated well and incubated for 3 h with CaCo-2 cells, while 1 mL of DMEM medium was added to the control wells. After washing four times with prewarmed PBS (pH 7.4), the cells were treated with 1 mL of 1% Triton X-100 (Sangon Biotech, Shanghai, China) for 10 min. Finally, the number of viable cell-adhering *L. rhamnosus* was serially diluted for plate counts and incubated at 37 °C for 48 h. The results were expressed as a ratio of the number of adherent bacteria to the average number of CaCo-2 cells in each well [31].

#### 2.6.6. Tolerance to Hydrogen Peroxide

Samples (100 µL) of *L. rhamnosus* X253 or *L. rhamnosus* GG bacterial suspensions (10^8^ CFU/mL) were seeded in 10 mL of MRS broth with different H_2_O_2_ concentrations (0, 1, 2, 3, 4, and 5 mM) [36], incubated at 37 °C for 24 h, and the absorbance at OD_600_ was measured every 2 h.

#### 2.6.7. In Vitro Assessment of Antioxidant Activity

Scavenging activity against 2,2-diphenyl-1-picrylhydrazyl free radicals: The 2,2-diphenyl-1-picrylhydrazyl (DPPH) free-radical-scavenging activity of *L. rhamnosus* X253 or *L. rhamnosus* GG was assessed according to the method described by Chen et al. [37]. Three-milliliter bacterial suspensions with different densities (10^4^, 10^6^, and 10^8^ CFU/mL) were mixed with one milliliter of ethanolic DPPH solution (0.2 mM). The mixtures were incubated in the dark at room temperature for 30 min. DPPH solution was mixed with sterilized water as a control group and incubated under the same conditions. The absorbance was measured at a wavelength of 517 nm, and the scavenging ability was calculated according to the following formula:(4)$$\mathrm{Scavenging}\;\mathrm{ability}\;(\%)=\frac{A_{\mathrm{control}}-A_{\mathrm{sample}}}{A_{\mathrm{control}}}\;\times \;100\%$$

Scavenging activity against hydroxyl radicals: The hydroxyl radical scavenging activity of *L. rhamnosus* X253 or *L. rhamnosus* GG was determined by the method of Tang [36], with some modifications. A reaction mixture containing 0.5 mL of ethanolic salicylic acid solution (5 mM), 0.5 mL of FeSO_4_ (5 mM), and 0.5 mL of H_2_O_2_ (3 mM) was prepared. The mixture was combined with 3 mL of bacterial suspension with different densities (10^4^, 10^6^, and 10^8^ CFU/mL) and adjusted to 5 mL with sterile water, before being incubated at 37 °C for 30 min. After centrifugation at 6000 g for 10 min, the absorbance of the supernatant was measured at 510 nm. The control group received an equal volume of sterilized water instead of the bacterial suspension. The hydroxyl radical scavenging activity was calculated according to the following formula:(5)$$\mathrm{Scavenging}\;\mathrm{ability}\;(\%)=\frac{A_{\mathrm{control}}-A_{\mathrm{sample}}}{A_{\mathrm{control}}}\;\times \;100\%$$

### 2.7. Statistical Analysis

Data from triplicate parallel experiments were used to calculate means and standard deviations. Statistical significance between groups was evaluated using Student’s *t* test.

## 3. Results and Discussion

### 3.1. Genome Properties

The whole genome of *L. rhamnosus* X253 was sequenced, assembled, and submitted to NCBI under the accession number CP073711. The genomic features of strain X253 were compared with those of the other nine typical *L. rhamnosus* strains from the two sources—dairy and human intestine (Table 2)—including the commercial strains Lc 705, HN001, R0011, and GG. The average genome size of the 10 strains was approximately 3.0 Mb, which matched the results reported by Lebeer et al. [38]. The number of encoded proteins and the GC content of strain X253 were similar to those of the other nine *L. rhamnosus* strains. Strain X253 encoded 15 rRNA genes, which was comparable to most other strains, except for HN001 and WQ2, which had only 3 and 2 rRNA genes, respectively. Moreover, *L. rhamnosus* X253 and strains from the human intestine were devoid of plasmids, unlike the other four strains derived from dairy.

### 3.2. Phylogenetic Analyses

The pan-genomes of the 10 *L. rhamnosus* strains contained 4427 genes, among which 1958 genes were core genes that were shared by the 10 strains (Figure 1A). With 20 unique genes in total, *L. rhamnosus* X253 had the lowest number of unique genes compared to the other nine strains. These unique genes encode the ABC transporter and ferrous iron transport protein b. The role of these unique genes for strain X253 needs further investigation.

The phylogenetic tree of the 10 *L. rhamnosus* strains was constructed based on 1936 single-copy orthologous genes (SOGs) to provide insight into their evolutionary connections and niche adaptations. According to the results, strains originating from dairy and the human intestine were neither completely separated nor clustered, which was consistent with the report by Jeong et al. [39]. *L. rhamnosus* X253 was more closely related to *L. rhamnosus* LOCK908 than *L. rhamnosus* GG (Figure 1B).

Average nucleotide identity (ANI) is the conventional criterion for evaluating whether or not two strains belong to the same species, and 95% is often considered to be the species threshold [40]. The ANI values between all 10 strains exceeded 97%. Consistent with the SOG-based phylogeny, whole-genome ANI analysis revealed that *L. rhamnosus* X253 was more closely related to *L. rhamnosus* LOCK908 (ANI approximately 100%) and *L. rhamnosus* Lc 705 (ANI 99.92%) than to *L. rhamnosus* GG (ANI 97.55%) (Figure 1C). The clustering of strains mirrored the phylogenetic tree and had no significant correlation with their origins of isolation.

Even though all 10 strains shared a general genomic similarity, a comparison of strain X253 and strain GG revealed specific genetic variations between the two that seemed essential for their niche adaptations. As shown in Figure 1D, there were multiple structural variants between the two genomes of X253 and GG. In comparison to the GG genome, the X253 genome contained a region with complex insertions and deletions between 0.3 and 0.6 Mb. This region comprised the *spa*CBA pilin gene cluster with a mucus-binding LPXTG motif, which might be important for strain GG to colonize in the human intestinal tract [9]. *Spa*CBA pilin gene clusters were absent in all dairy-derived isolates [9], which might affect their colonizing capability. Additionally, the X253 genome had four deletions longer than 5000 bp, which were primarily composed of genes related to phagosomes, transposases, recombinases, and carbohydrate metabolism (particularly fucose metabolism). In three insertion sequences larger than 5000 bp in the X253 genome, genes related to cellular SOS response to DNA damage were identified, such as S24 family peptidase (KEM50_RS03990) and LexA family transcriptional regulator (KEM50_RS03995) [41].

### 3.3. Comparative Functional Analysis between the X253 and GG Genomes

The predicted functional capacities of strain X253 and GG were further investigated in relation to the genomes of the eight other *L. rhamnosus* strains using the results from the COG functional annotation (Figure 2). For several categories—such as coenzyme transport and metabolism (category H); translation, ribosome structure, and biogenic function (category J); and cell motility, intracellular trafficking, secretion, and vesicular transport (category N)—strains X253 and GG were positioned close to one another, suggesting that these categories were functionally conserved between the two strains. On the other hand, for the categories of carbohydrate transport and metabolism (category G); transcription (category K); replication, recombination, and repair (category L); cell wall, membrane, and envelope biogenesis (category M); and defense mechanisms (category V), strains X253 and GG were far apart. These results suggest that X253 and GG might be different in terms of their carbohydrate usage, and they may have evolved to adapt to different niches by adopting distinct transcriptional, repair, and other regulatory mechanisms. Consequently, the results provide a reference for comparing the functional differences between the two strains.

### 3.4. Safety Evaluation of L. rhamnosus X253

#### 3.4.1. Antibiotic Resistance

Antibiotic resistance in bacteria poses a serious public health threat. Bacteria might acquire resistance to antibiotics by horizontal transfer of plasmids, foreign DNA recombination, or mutations at chromosomal locations [14]. Having no plasmid in its genome, *L. rhamnosus* X253 might have lower risk of transmitting antibiotic resistance. Similar to strain GG, only one GE2270A resistance gene was identified for strain X253 in the CARD database with coverage >60% and identity >75% (Appendix A). As a thiopeptide isolated from a strain of *Planobispora rosea*, GE2270A binds to elongation factor thermo unstable (EF-Tu) and blocks the delivery of aminoacyl tRNAs to the ribosome [42]. The antibacterial activity of GE2270A has been demonstrated in vitro against *Staphylococcus aureus*, *Streptococcus pyogenes*, *Enterococcus faecalis*, *Clostridium perfringens*, and *Propionibacterium acnes*, with minimum inhibitory concentrations below 1 μg/mL [42,43]. In this study, GE2270A at 0.5 μg/mL significantly hampered the growth of *L. rhamnosus* X253 and *L. rhamnosus* GG. Due to the lack of a criterion to determine whether or not a strain is resistant to GE2270A, it was impossible to accurately judge the resistance of these two strains.

As mentioned in the EFSA guidance document, for *L. rhamnosus* strains intended for human and veterinary use as feed additives, their susceptibility to a certain spectrum of antibiotics needs to be evaluated [32]. The MIC results showed that both *L. rhamnosus* X253 and GG were sensitive to ampicillin, erythromycin, gentamicin, streptomycin, and tetracycline. However, both strains were resistant to vancomycin. As Korhonen et al. reported, *L. rhamnosus* lacked vancomycin binding due to the replacement of D-alanine residues with D-lactic acid at the pentapeptide end of the cell wall [44]. In addition, the presence of the ABC antibiotic efflux pump for chloramphenicol in both strains might have resulted in the reduction in the intracellular concentration of chloramphenicol, thereby increasing the resistance of both strains to chloramphenicol (Table 3), and the results were consistent with those of previous studies [9,45].

#### 3.4.2. Virulence Factor Genes and Toxin-Encoding Genes

In the VFBD database, 215 and 212 virulence genes were predicted for *L. rhamnosus* X253 and GG, respectively, but all of these genes displayed low similarity, with an identity of less than 75% (Appendix A). As with *L. rhamnosus* GG, the genome of X253 did not encode toxic factors, including hemolysin BL (*Hbl*), non-hemolytic enterotoxin (*Nhe*), and enterotoxin (*cytotoxin K*). The hemolytic activity of both strains was further confirmed by phenotypic analysis. When cultured on Columbia Blood Agar Plates containing fresh sheep blood for 48 h, neither *L. rhamnosus* X253 nor GG exhibited hemolytic activity, whereas *S. aureus* ATCC 6538 demonstrated significant β-hemolytic activity (Figure 3).

#### 3.4.3. Enzymatic Profile by API ZYM

*L. rhamnosus* X253 and GG both contained the gene *uid*A encoding β-glucuronidase. β-Glucuronidase is a potentially carcinogenic enzyme that increases the risk of gastric cancer and inflammatory bowel disease [46]. Therefore, it was necessary to analyze the extracellular enzyme profiles of both strains phenotypically. The enzyme profile of *L. rhamnosus* X253 was similar to that of *L. rhamnosus* GG (Table 4). *L. rhamnosus* X253 and GG produced several non-hazardous enzymes, including esterase, leucine arylamidase, valine arylamidase, cystine arylamidase, acid phosphatase, and naphthol-AS-BI-phosphohydrolase. No β-glucuronidase activity was found, indicating that both strains are safe.

### 3.5. Assessment of Probiotic Properties

#### 3.5.1. Carbohydrate Fermentation Profiles

As the genes involved in carbohydrate transport and metabolism were considerably different between *L. rhamnosus* X253 and *L. rhamnosus* GG, phenotypic analysis was conducted to determine the differences between the two strains with regard to carbohydrate utilization. The carbohydrate usage profiles revealed that *L. rhamnosus* X253 and GG were able to use 23 and 19 carbohydrates, respectively. Utilization of carbohydrates is dependent on functional transporters and intact metabolic pathways. Discrepancies between the two strains may result from the gain and loss of metabolism-related genes in their evolution in distinct niches. Strain GG was capable of using D-arabinose, dulcitol, and L-fucose, whereas strain X253 did not possess these abilities but could ferment L-sorbose, L-rhamnose, methyl-α-D-glucopyranoside, D-maltose, D-lactose, D-saccharose, and D-turanose (Table 5).

Genetic differences between the two strains may explain some of their differences. The genome of *L. rhamnosus* GG contains a tagatose-6-phosphate pathway (*lac*ABCD) and a lactose PTS (*lac*FEG); however, the antiterminator (*lac*T) and the phospho-β-galactosidase (*lac*G) genes are mutated and inactive, making GG incapable of metabolizing D-lactose [7]. In strain X253, the *lac*T and *lac*G genes were conserved, consistent with the fact that lactose is an important carbohydrate in dairy products. In contrast, *L. rhamnosus* GG utilized L-fucose better than *L. rhamnosus* X253. Moreover, consistent with the fact that strain X253 cannot use L-fucose, it did not encode the *fcs*R fucose operon repressor, *fuc*U, or *fuc*I isomerase [47]. As an important carbon source in the intestine, L-fucose can be found in a variety of fucosylated compounds, including human mucins and glycoproteins [47].

#### 3.5.2. Artificial Gastric Fluid and Intestinal Fluid Tolerance Assays

*L. rhamnosus* GG is a typical strain with a high tolerance to the conditions in the gastrointestinal tract [8]. The survivability of *L. rhamnosus* X253 and *L. rhamnosus* GG in artificial gastric fluids (AGF) and artificial intestinal fluids (AIF) is shown in Figure 4. The survival rates of both *L. rhamnosus* X253 and GG remained above 90% when incubated in AGF (pH 3.0) for 3 h. Genes encoding proteins that are involved in acid tolerance—such as dTDP-glucose 4,6-dehydratase [48], 4-hydroxy-tetrahydrodipicolinate synthase, ATP-dependent intracellular protein ClpP [49], and F0F1-ATPases (such as *atp*C, *atp*D, *atp*G, *atp*A, *atp*H, *atp*F, and *atp*B) [50]—were found in the genomes of both strains. These genes might contribute to the adaptation of *L. rhamnosus* X253 and GG in low-pH environments.

Despite this, the survival rate of GG was maintained around 80% after a further 5 h of AIF (pH 8.0) stress, whereas the survival rate of X253 declined dramatically to approximately 55%. This difference might be related to the fact that *L. rhamnosus* GG possesses genes encoding taurine transport systems (*tau*ABC), but *L. rhamnosus* X253 does not. These genes may be involved in the binding of bile salts [51], allowing GG to tolerate bile salts better than X253.

#### 3.5.3. Adhesion Ability

The hydrophobicity and auto-aggregation of bacteria are indicators of the non-specific adhesion of probiotics to the intestine [52]. The hydrophobicity and auto-aggregation capacity of *L. rhamnosus* X253 were 51.43 ± 3.49% and 76.08 ± 3.23%, respectively, and these properties were significantly lower than those of GG (Figure 5). Using CaCo-2 cells, the adhesion of *L. rhamnosus* X253 and GG was further evaluated. The results showed that each CaCo-2 cell was attached by an average of 9.57 ± 0.87 GG cells, which was significantly higher than that of strain X253. The *spa*CBA pilin gene cluster is not commonly found in dairy-derived strains of *L. rhamnosus* [9], which perhaps explains why strain X253 had a lower adhesion capability than strain GG. Moreover, the adhesion embodied in strain X253 may be linked to other genes, such as the molecular chaperones *dnaK* and *groEL* [53].

#### 3.5.4. Hydrogen Peroxide Tolerance Assay

By adding different concentrations of H_2_O_2_ to MRS broth, the tolerance of *L. rhamnosus* X253 and GG to H_2_O_2_ was determined. The results demonstrated that the lag period of both strains increased with increasing H_2_O_2_ concentrations, possibly owing to the oxidative damage caused to the bacteria by the high H_2_O_2_ concentration. In addition, *L. rhamnosus* X253 was able to survive at a concentration of 3 mM H_2_O_2_—higher than GG could handle (Figure 6A,B), and also much higher than other *L. rhamnosus* strains, as previously reported [29].

#### 3.5.5. Antioxidant Activity In Vitro

Certain metabolites generated by probiotics have been demonstrated to decrease oxidative damage, making them helpful in delaying aging and preventing chronic illness [54]. Studies have confirmed that *L. rhamnosus* GG can ameliorate alcohol-induced oxidative damage in the murine intestine and boost antioxidant indicators in the human body [55,56]. The rates of scavenging of DPPH and hydroxyl radicals by bacterial suspensions of strains X253 and GG revealed that the antioxidant capacity of both strains was proportional to the bacterial densities. Strains X253 and GG both had good antioxidant capacity, which might have been due to the presence of antioxidant stress genes in their genomes, such as thioredoxin, glutathione peroxidase, pyruvate oxidase, and NADH-dependent flavin oxidoreductase. Notably, strain X253 demonstrated higher levels of DPPH scavenging ability than strain GG. In addition, strain X253 was substantially more effective than GG at 10^8^ CFU/mL in scavenging hydroxyl free radicals, with rates of 30.21 ± 1.75% and 26.08 ± 1.21%, respectively. According to these results, strain X253 had greater antioxidant activity than strain GG, suggesting its potential as a disease-preventive and anti-aging agent [57,58].

## 4. Conclusions

A comparative genomic analysis of 10 *L. rhamnosus* strains revealed that the dairy-derived strain X253 was most closely linked to the human-intestine-derived strain LOCK908, and the clustering pattern did not correspond well to the strains’ origin of isolation. The genetic differences between strain X253 and strain GG enable them to occupy different niches. Using genetic analysis and phenotypic validation, strain X253 was confirmed to be comparable to strain GG in terms of safety. The difference in probiotic characteristics between strain X253 and strain GG could also be attributed to their origins of isolation. Notably, strain X253 demonstrated superior hydrogen peroxide tolerance and antioxidant activity compared to strain GG. In summary, this study on the phenotypic and genetic aspects of *L. rhamnosus* X253 confirmed its safety and probiotic properties, making it a promising probiotic candidate.

## Figures and Tables

**Figure 1 microorganisms-11-00140-f001:**
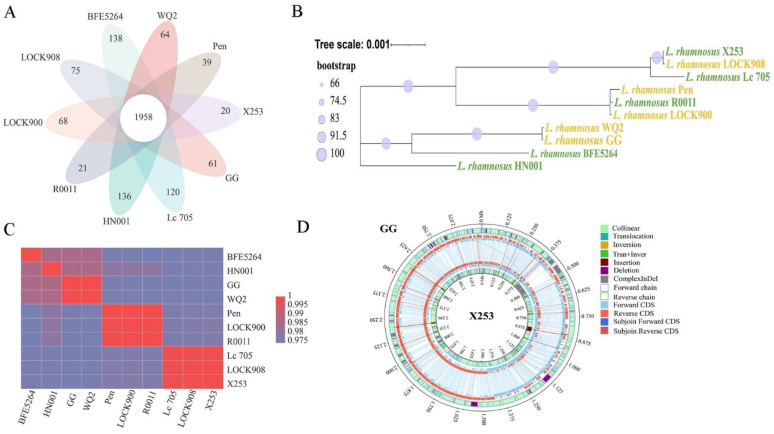
Comparative genomic analysis of *L. rhamnosus* X253 with other *L. rhamnosus* strains: (**A**) Venn diagram illustrating the number of genes in the core genome and unique genes of the pan-genome of 10 *L. rhamnosus* strains. (**B**) Phylogenetic tree constructed on single-copy orthologous genes to reveal the genetic distance of 10 *L. rhamnosus* strains. (**C**) Heatmap of ANI analysis of the genomes of 10 *L. rhamnosus* strains. (**D**) Genomic comparison between *L. rhamnosus* X253 and *L. rhamnosus* GG.

**Figure 2 microorganisms-11-00140-f002:**
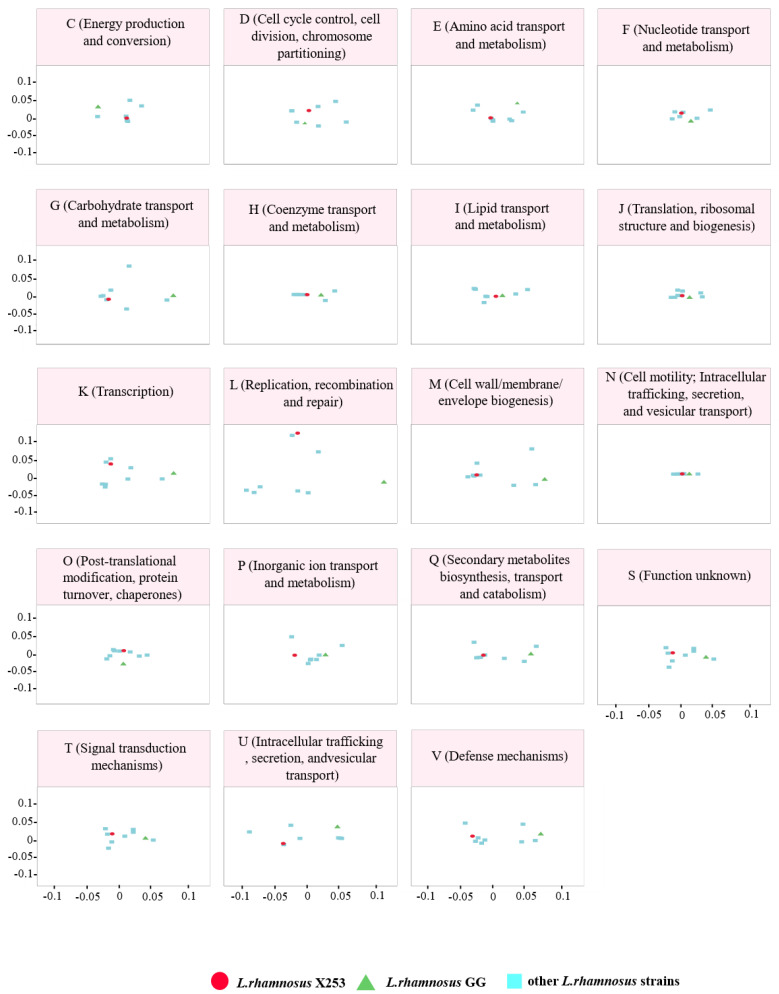
PCoA of the predicted functional capacity of 10 *L. rhamnosus* strains based on mapping of all orthogroups to the eggNOG database. Distance matrices based on orthogroup abundance profiles were calculated with the R package vegan (version 2.4.4) and visualized with ggplot2 (version 3.3.5). Each letter represents a different functional category, and *L. rhamnosus* X253, *L. rhamnosus* GG, and the other eight strains are denoted by red circles, green triangles, and blue squares, respectively.

**Figure 3 microorganisms-11-00140-f003:**
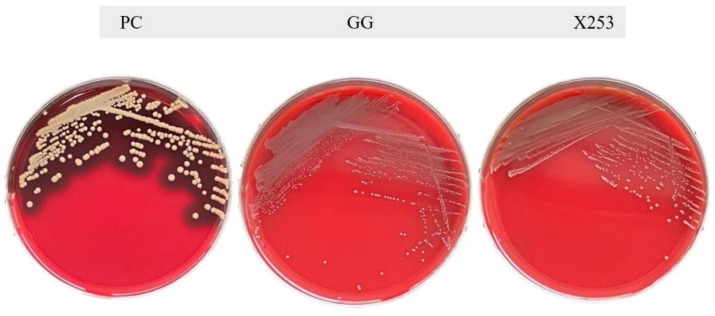
Hemolytic ability of *L. rhamnosus* GG and X253: As a positive control, *S. aureus* ATCC 6538 generated a clearly visible zone of β-hemolytic activity.

**Figure 4 microorganisms-11-00140-f004:**
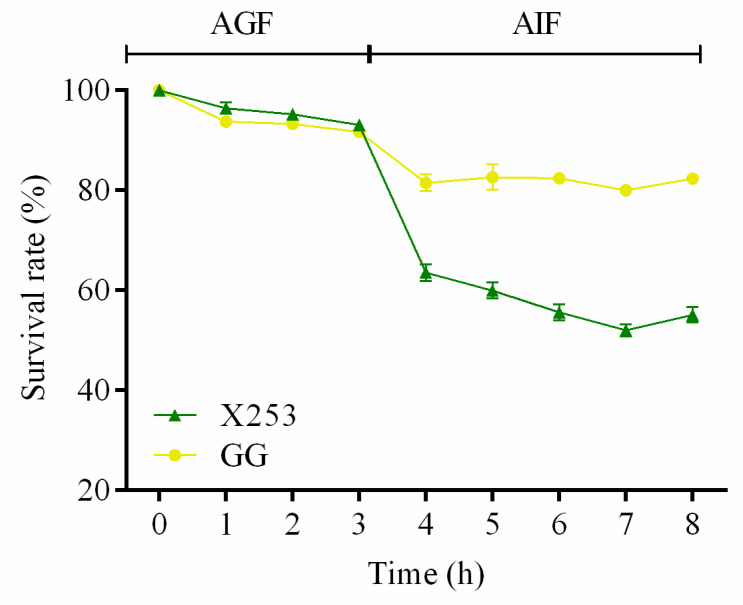
Artificial gastric fluid and intestinal fluid tolerance of *L. rhamnosus* X253 and GG: Survival rate of *L. rhamnosus* X253 and GG (10^8^ CFU/mL) in artificial gastric fluids at pH 3.0, followed by artificial intestinal fluids at pH 8.0.

**Figure 5 microorganisms-11-00140-f005:**
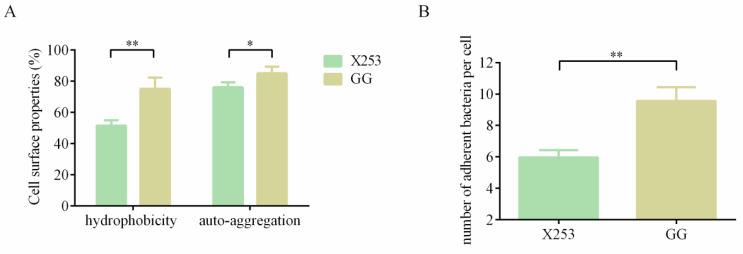
Comparative analysis of *L. rhamnosus* X253 and *L. rhamnosus* GG for hydrophobicity, auto-aggregation and adhesion to CaCo-2 cells; *: *p* < 0.05, **: *p* < 0.01. (**A**) Hydrophobicity and auto-aggregation. (**B**) Adhesion to CaCo-2 cells.

**Figure 6 microorganisms-11-00140-f006:**
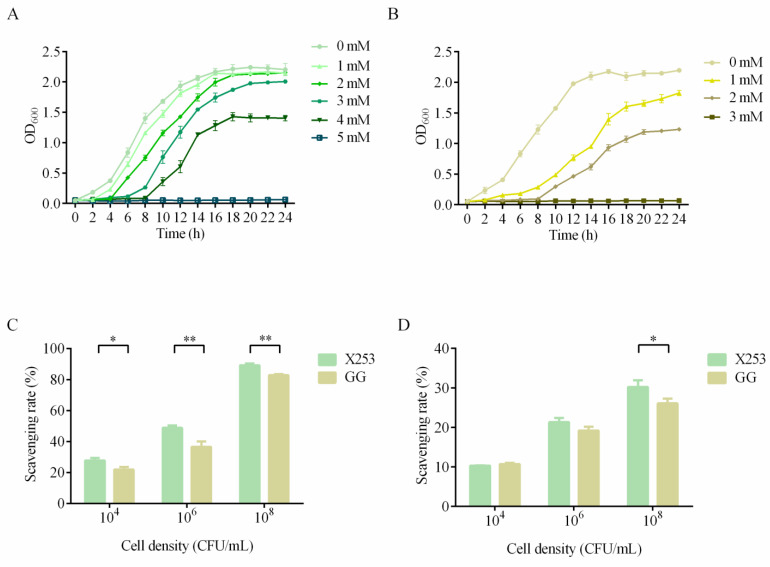
Comparative analysis of hydrogen peroxide tolerance and antioxidant capacity of *L. rhamnosus* X253 and *L. rhamnosus* GG: (**A**,**B**) Growth of *L. rhamnosus* X253 (**A**) and *L. rhamnosus* GG (**B**) in MRS broth containing different concentrations of hydrogen peroxide. (**C**,**D**) Scavenging rates of DPPH (**C**) and hydroxyl radicals (**D**) by bacterial suspensions with different densities (10^4^, 10^6^, and 10^8^ CFU/mL) of strains X253 and GG. *: *p* < 0.05, **: *p* < 0.01.

**Table 1 microorganisms-11-00140-t001:** *L. rhamnosus* strains used in this study.

Species	Strain	Assembly No.	Source
*L. rhamnosus*	X253	GCA_018228745.1	Dairy
	BFE5264	GCA_001988935.1	
	Lc 705	GCA_000026525.1	
	R0011	GCA_000235785.2	
	HN001	GCA_000173255.2	
	Pen	GCA_002076955.1	Human intestine
	GG	GCA_000026505.1	
	LOCK908	GCA_000418495.1	
	LOCK900	GCA_000418475.1	
	WQ2	GCA_002025085.1	

**Table 2 microorganisms-11-00140-t002:** Genome features of 10 representative *L. rhamnosus* genomes.

Strain	Size (Mb)	Plasmids	CDS	GC%	rRNAs	tRNAs	Source
X253	2.99	0	2649	46.8	15	62	Dairy
BFE5264	3.11	1	2785	46.8	15	60	
Lc 705	3.03	1	2652	46.6	15	61	
R0011	2.90	N/A	2618	46.7	11	54	
HN001	2.91	2	2642	46.7	3	50	
Pen	2.88	0	2569	46.8	15	59	Human intestine
GG	3.01	0	2703	46.7	15	57	
LOCK908	2.99	0	2666	46.8	15	62	
LOCK900	2.88	0	2586	46.8	15	59	
WQ2	2.96	0	2653	46.7	2	45	

N/A: the draft genome sequence of R0011 consists of 10 contigs, and no plasmids were identified.

**Table 3 microorganisms-11-00140-t003:** MIC values observed for *L. rhamnosus* X253 and GG against several antibiotics.

Antibiotic	MIC Cut-Off Values (μg/mL)	MIC Observed (μg/mL)		Results	
		X253	GG	X253	GG
Ampicillin	4	0.5	0.5	S	S
Chloramphenicol	4	8	8	R	R
Erythromycin	1	0.5	0.5	S	S
Gentamicin	16	16	8	S	S
Streptomycin	32	8	16	S	S
Tetracycline	8	0.5	1	S	S
Vancomycin	N/R	>256	>256	R	R

S: sensitive, R: resistant.

**Table 4 microorganisms-11-00140-t004:** Enzymatic profiles and assay of toxic metabolic production.

Substrate	X253	GG
Alkaline phosphatase	+	+
Esterase (C4)	+	+
Esterase lipase (C8)	+	+
Lipase (C14)	−	−
Leucine arylamidase	+	+
Valine arylamidase	+	+
Cystine arylamidase	+	+
Trypsin	−	−
α-Chymotrypsin	+	+
Acid phosphatase	+	+
Naphthol-AS-BI-phosphohydrolase	+	+
α-Galactosidase	−	−
β-Galactosidase	+	+
β-Glucuronidase	−	−
α-Glucosidase	+	+
β-Glucosidase	+	+
N-acetyl-glucosaminidase	−	−
α-Mannosidase	−	−
β-Fucosidase	+	+

+: positive, −: negative.

**Table 5 microorganisms-11-00140-t005:** Carbohydrate fermentation profile assessed by API 50 CHL test strips.

Substrate	X253	GG	Substrate	X253	GG
Glycerol	−	−	Esculine	+	+
Erythritol	−	−	Salicine	+	+
D-arabinose	−	+	D-cellobiose	+	+
L-arabinose	−	−	D-maltose	+	−
D-ribose	−	−	D-lactose	+	−
D-xylose	+	−	D-melibiose	−	−
L-xylose	−	−	D-saccharose	+	−
D-adonitol	−	−	D-trehalose	+	+
Methyl-β-D-xylopyranoside	−	−	Inulin	−	−
D-galactose	+	+	D-melezitose	+	+
D-glucose	+	+	D-raffinose	−	−
D-fructose	+	+	Starch	−	−
D-mannose	+	+	Glycogen	−	−
L-sorbose	+	−	Xylitol	−	−
L-rhamnose	+	−	Gentiobiose	−	+
Dulcitol	−	+	D-turanose	+	−
Inositol	−	−	D-lyxose	−	−
D-mannitol	+	+	D-tagatose	+	+
D-sorbitol	+	+	D-fucose	−	−
Methyl-α-D-mannopyranoside	−	−	L-fucose	−	+
Methyl-α-D-glucopyranoside	+	−	D-arabitol	−	−
N-acetylglucosamine	+	+	L-arabitol	−	−
Amygdaline	+	+	Potassium gluconate	−	−
Arbutine	+	+	Potassium 2-cetogluconate	−	−

+: positive, −: negative.

## Data Availability

The full genome data of *L. rhamnosus* X253 can be found in NCBI GenBank under the accession number CP073711.

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
