# Peer review of "Assessing the Safety and Probiotic Characteristics of Lacticaseibacillus rhamnosus X253 via Complete Genome and Phenotype Analysis"

_microorganisms, 2023, doi:10.3390/microorganisms11010140_

Round 1

Reviewer 1 Report

The manuscript of Zhao et al. reported the full genome sequences of Lacticaseibacillus rhamnosus X253, the phylogenic relation of the new strain to other strains of the same species, and some genomic and phenotypic features of it concerning its safety and probiotic properties. It provided some useful information for our understanding the probiotic bacteria. There are some concerns that need to be addressed.

 1) In Fig.1C, strain HN001 and X253 were situated very closely, give readers an impression that they were closely related. But that was not the case (Fig 1B, ANI result). Fig.1 C should be re-do to prevent the mis-understanding. In addition, the letters in Fig 1C and D were too small to read.

2) The authors discussed that there were deletions and insertion when comparing genomes of X253 and GG in the test. These sites should be indicated in the figure (Fig.1D).

3) The authors should describe briefly the meaning of Fig.2 and how the distance between strains was obtained. 

4) There are several grammatical errors in the manuscript. It need to be carefully rewrite make the manuscript more concise and accurate. 

Author Response

Response to Reviewer 1 Comments

Point 1: In Fig.1C, strain HN001 and X253 were situated very closely, give readers an impression that they were closely related. But that was not the case (Fig 1B, ANI result). Fig.1 C should be re-do to prevent the mis-understanding. In addition, the letters in Fig 1C and D were too small to read. 

Response: Thanks for the comment. In new Figure 1B, we have modified the circular evolutionary tree to a rectangular version in order to make the relationship between the ten L. rhamnosus strains more visible. Moreover, we have made all the letters in Figure 1 larger in order to improve their legibility.

Point 2: The authors discussed that there were deletions and insertion when comparing genomes of X253 and GG in the test. These sites should be indicated in the figure (Fig.1D).

Response: Thank you for your suggestion. We have re-done Figure 1. The inner circle represents the genome of X253 and the outer circle represents the genome of GG. The Collinear, Translocation, Inversion, Insertion, Deletion and ComplexInDel regions of the two genomes are marked on the diagram. The ComplexInDel indicates a region where two genomes do not match but are positioned opposite each other.

Point 3: The authors should describe briefly the meaning of Fig.2 and how the distance between strains was obtained. 

Response: Thank you for your suggestion. The analysis in Figure 2 refers to the study by Petrova et al., we have added some sentences to describe how the distance between strains was obtained. “Additionally, the functional capacity of ten strains was investigated by matching the genomic sequences to the eggNOG database (version 5.0, http://eggnogdb.embl.de/#/app/home, accessed on 22 June 2022). Distance matrices based on orthogroup abundance profiles were calculated with the R package vegan (version 2.4.4) and visualized with ggplot2 (version 3.3.5).” (Lines 232-236 on page 3).

In addition, the legend of Figure 2 has been amended to clarify its meaning. “PCoA of predicted functional capacity of ten L. rhamnosus strains based on mapping of all orthogroups to the eggNOG database. Distance matrices based on orthogroup abundance profiles were calculated as described by Petrova et al [29]. Each letter represented a different functional category, and L. rhamnosus X253, L. rhamnosus GG and other eight strains were denoted by red circles, green triangles and blue squares respectively.” (Lines 737-739 on page 9).

Point 4: There are several grammatical errors in the manuscript. It needs to be carefully rewrite make the manuscript more concise and accurate. 

Response: Thanks for the comment. We have carefully checked the English language of the manuscript, and made many corrections and modifications.

Reviewer 2 Report

In this work, comparative genomic and phenotypic analyses were performed to assess the safety characteristics and probiotic traits of L. rhamnosus X253 derived from fermented milk in China and compared with various L. rhamnosus strains derived from the human intestine. This work is a very interesting and well-presented study. I have a simple question.

In Fig 3, the hemolytic ability of LGG and X253 was compared with S. aureus ATCC 6538 as a positive control. Toxin genes such as hemolysin BL (Hbl), non-hemolytic enterotoxin (Nhe), and enterotoxin (cytotoxin K) described by the author in lines 363-366 are major toxin genes of the Bacillus cereus group strains. 

S. aureus also has some hemolytic genes. Is there a specific reason for referring to the hemolytic genes of Bacillus cereus instead of S. aureus in the text?

Author Response

Response to Reviewer 2 Comments

Point 1: In Fig 3, the hemolytic ability of LGG and X253 was compared with S. aureus ATCC 6538 as a positive control. Toxin genes such as hemolysin BL (Hbl), non-hemolytic enterotoxin (Nhe), and enterotoxin (cytotoxin K) described by the author in lines 363-366 are major toxin genes of the Bacillus cereus group strains. S. aureus also has some hemolytic genes. Is there a specific reason for referring to the hemolytic genes of Bacillus cereus instead of S. aureus in the text?

Response: Thanks for the comment. We currently have only one strain of Staphylococcus aureus ATCC 6538 with β-hemolytic activity and used it as a positive control to demonstrate the non-β-hemolytic activity of L. rhamnosus X253 and L. rhamnosus GG. Blast analysis revealed that not only were hemolysin BL (Hbl), non-hemolytic enterotoxin (Nhe), and enterotoxin (cytotoxin K) absent from the genomes of both strains, but also virulence genes of S. aureus, such as exfoliative toxin (eta, etb), staphylococcal enterotoxin (sea, seb, sec1, sec3, sed, see, seh, selk, selq).
